# Mannitol Is a Good Anticaking Agent for Spray-Dried Hydroxypropyl-Beta-Cyclodextrin Microcapsules

**DOI:** 10.3390/molecules28031119

**Published:** 2023-01-22

**Authors:** Xingran Kou, Xinping Zhang, Ying Cheng, Miao Yu, Qingran Meng, Qinfei Ke

**Affiliations:** 1Collaborative Innovation Center of Fragrance Flavor and Cosmetics, School of Perfume and Aroma Technology, Shanghai Research Institute of Fragrance & Flavour Industry, Shanghai Institute of Technology, Shanghai 201418, China; 2Key Laboratory of Textile Science & Technology, Ministry of Education, College of Textiles, Donghua University, Shanghai 200051, China

**Keywords:** spray drying, microcapsules, mannitol, crystallisation, caking

## Abstract

Agglomeration is an undesirable phenomenon that often occurs in spray-dried microcapsules powder. The objective of this work is to determine the best solution for spray-dried hydroxypropyl-β-cyclodextrin (HP-β-CD) microcapsules from four anticaking agents, namely calcium stearate (CaSt), magnesium stearate (MgSt), silicon dioxide (SiO_2_), and mannitol (MAN), and to explore their anticaking mechanisms. Our results showed that MAN was found to be the superior anticaking agent among those tested. When the MAN ratio is 12%, the microcapsules with a special Xanthium-type shape had higher powder flowability and lower hygroscopicity and exhibited good anticaking properties. Mechanism research revealed that CaSt, MgSt, and SiO_2_ reduce hygroscopicity and caking by increasing the glass transition temperature of the microcapsules, while MAN prevents the hydroxyl group of HP-β-CD from combining with water molecules in the air by a crystal outer-layer on the microcapsule surface.

## 1. Introduction

Spray drying is a commonly used technique for powdering microcapsules [1]. It is a well-studied method that has been applied in the fields of food, flavour and fragrance, pharmaceuticals, chemicals, coatings, and dyes [2,3,4,5]. Spray drying comprises the rapid evaporation of a solvent from atomised droplets, forming dry powder or particles. The solution is atomised in an atomiser into many tiny droplets before entering a dryer, in which the droplets contact hot air, i.e., the drying medium. Heat transfer from the hot air raises the temperature of the droplets and the solvent starts to volatilise [6]; consequently, parts of the solids precipitate on the surface of the droplets, forming a dry crust on the surface that hinders the mass transfer process [7,8]. Subsequently, the vaporised solvent migrates to the surface of the microcapsule shell and volatilises, thereby forming a dry microcapsule powder. In the spray-drying process, the atomisation and drying stages have the most influence on the powder properties.

Microencapsulated powders are often agglomerated during spray drying [9]. Moreover, depending on the nature of the material and the spray-drying process, excessive agglomeration of the microcapsules may occur [10], which is detrimental to the drying and storage of microcapsules. This unwanted natural phenomenon is typically called caking, and the development of microcapsules without a tendency to cake is an ongoing challenge.

Microencapsulated powders are divided into crystalline and amorphous forms, and the mechanism of agglomeration differs between these forms. The agglomeration of crystalline solids is mainly due to liquid bridging and capillary action, resulting in the ‘hard agglomeration’ of microcapsules, whereby the microcapsules continually harden during the agglomeration process. Amorphous solids have a higher solubility and dissolution rate than crystalline solids but are not as physically and chemically stable; thus, the agglomeration of amorphous solids is mainly due to ‘soft agglomeration’ caused by the glass transition [11,12], with the microcapsules absorbing water leading to the plasticisation of the particle surface. When the surface viscosity drops to a critical value (10^6^–10^8^ Pas), the adjacent particles form bridges and agglomerate [13].

The use of anticaking agents, such as silicates, stearates, and polysaccharides, can improve the flowability and anticaking ability of microcapsules. The effects of different anticaking agents are due to different mechanisms. For example, microcapsule particles are completely covered by the anticaking agent, which reduces the friction between microcapsules through a physical barrier effect. In addition, the anticaking agent may compete with the microcapsule materials for moisture absorption, thereby reducing the tendency of the microcapsules to cake because of hygroscopicity. The addition of anticaking agents such as calcium stearate (CaSt) and silica (SiO_2_) can reduce moisture content and increase the glass transition temperature, thereby improving the anticaking resistance of spray-dried microcapsules [14]. Mannitol (MAN), a sugar alcohol with low hygroscopicity, is widely used as a confectionery tablet press owing to its high water solubility and low toxicity. However, the mechanism of MAN as an anticaking agent has not been studied in detail.

Anticaking agents are effective only if they are compatible with the characteristics of the microencapsulated materials. In the present study, oil-soluble lavender essence was used as the core material and hydroxypropyl-beta-cyclodextrin (HP-β-CD) was used as the wall material. HP-β-CD is a derivative of beta-cyclodextrin, which is non-toxic and non-hazardous. Its external hydrophilicity is strong, and the hydrophobic structure of its inner cavity can selectively load the hydrophobic essence in this cavity. Thus, HP-β-CD can enhance the stability of chemical properties, improve water solubility, and achieve slow release and volatility reduction; thus, it has a wide range of applications. However, HP-β-CD microcapsules prepared via spray drying exhibit agglomeration. The hydroxyl group of HP-β-CD combines easily with water molecules in air to form hydrogen bonds, leading to high moisture content. Indeed, the hygroscopicity of HP-β-CD is the cause of adhesion and caking in HP-β-CD microcapsules.

In the current study, various anticaking agents, including CaSt, MgSt, SiO_2_, and MAN were screened and evaluated for their anticaking effects with the aim of inhibiting the caking phenomenon in HP-β-CD microcapsules. The particle size distribution, morphology, moisture content, powder mobility, hygroscopicity, amorphization, and thermal stability of the microcapsules were evaluated. In addition, the effect of the MAN weight ratio on the morphology of HP-β-CD microcapsules was investigated with the aim of inhibiting the glass transition of the amorphous components and regulating the surface roughness and powder flowability of the microcapsules.

## 2. Results and Discussion

### 2.1. Morphology and Microstructure of Microcapsule Powders with Different Anticaking Agents

The intuitive particles of the various microcapsules are shown in Figure 1a. The HP-β-CD microcapsules without anticaking agent exhibited high levels of agglomeration and poor dispersion, whereas the microcapsules with anticaking agents exhibited different degrees of agglomeration. HP-β-CD/CaSt, HP-β-CD/MgSt, and HP-β-CD/SiO_2_ microcapsules exhibited poor anti-agglomeration properties, whereas HP-β-CD/MAN microcapsules exhibited no obvious agglomeration. HP-β-CD/MAN microcapsules took the form of fine flour, indicating that MAN had anti-agglomeration effects.

The surface morphology and internal structure of the microcapsules are shown in Figure 1b,c. The particle sizes of microcapsules in each system are distributed in the range of 1–20 μm. HP-β-CD microcapsules had many folds and depressions on their surface and few microcapsules were smooth and spherical, which was consistent with other microcapsules examined in previous studies [15,16]. This may be due to the rate at which the atomised droplets migrate to the surface being lower than the rate at which the surface moisture is vaporised at high temperature in the drying chamber. Under such conditions, the surface of the microcapsule quickly forms a hard shell, retaining moisture within the microcapsule [17]. The internal moisture then gradually migrates to the surface through the tiny pores of the microcapsule walls. When microcapsules collide, they appear to shrink owing to inward depression [18]. HP-β-CD/MAN microcapsules showed high sphericity, with some agglomeration of small particles around large particles. The formation of tiny agglomerates is beneficial because it increases the specific surface area of the microcapsules, thereby reducing the surface free energy. Simultaneously, the powder mobility is improved, and the formation of larger agglomerates is prevented [19]. Chew and Chan [20] found that particles with high roughness exhibited higher dispersibility than smooth particles. In the present study, the surface roughness of HP-β-CD/MAN microcapsules was higher that of other microcapsules. This was likely due to the increase in MAN and HP-β-CD concentrations as the moisture evaporated, with the high- and low-solubility-component HP-β-CD and MAN, respectively, tending to migrate to the inner and outer layer of the droplet, with the MAN forming crystals as its concentration becomes saturated [8,21,22]. This finding is consistent with the microcapsule crystallinity shown in Figure 2, explaining the rough morphology observed.

The HP-β-CD/CaSt and HP-β-CD/MgSt microcapsules exhibited slight wrinkles, whereas the surface morphology of the HP-β-CD/SiO_2_ microcapsule was similar to that of HP-β-CD microcapsules, i.e., many wrinkles and many small depressions were observed. This was likely due to SiO_2_ facilitating the formation of a harder ‘thin shell’ on the surface during the spray-drying process. With moisture migration, the shrinkage of the droplets was hindered, and fewer wrinkles appeared on the surface [23].

### 2.2. Particle Size Distribution, Moisture Content, and Hygroscopicity of Microcapsules with Different Anticaking Agents

#### 2.2.1. Particle Size Distribution of Microcapsules with Different Anticaking Agents

The particle sizes of the microcapsule emulsions ranged from 180.49 to 212.80 nm (Table 1), which was lower than that of HP-β-CD microcapsules. However, the particle sizes of microcapsules with anticaking agents were lower because the presence of the anticaking agent reduced microcapsule adhesion during the spray-drying process. Among the four microcapsule systems with anticaking agents, those with a small emulsion particle size had a small powder particle size owing to the faster heat exchange rate of the smaller emulsion droplets. The HP-β-CD/SiO_2_ microcapsule emulsion had the smallest particle size, and its spray-dried powder had the smallest volume-weighted mean diameter of 8.21 μm. In contrast, HP-β-CD/MAN microcapsule powder had a volume-weighted mean diameter of 11.30 μm, although the diameter of the HP-β-CD/MAN microcapsule was about 6 μm, as shown in observations under SEM (Figure 1). Because the HP-β-CD/MAN microcapsules adhered to each other, the particle sizes were larger when measured using a laser particle-size analyser. The appropriately low level of agglomeration in the microscopic state reduced the specific surface area and surface free energy of the microcapsules, which limited the caking of the powder in the macroscopic state and improved powder flowability.

#### 2.2.2. Moisture Content of Microcapsules with Different Anticaking Agents

The moisture content of microcapsules affects their fluidity and agglomeration. In general, microcapsules with a moisture content of 3–10% exhibit high stability during storage [24]. The microcapsules with different anticaking agents exhibited moisture contents of 3.43–4.55% (Table 1), among which the HP-β-CD microcapsule exhibited the highest moisture content, indicating that the anticaking agents reduced the moisture content of the microcapsules.

#### 2.2.3. Hygroscopicity of Microcapsules with Different Anticaking Agents

Microcapsules with low hygroscopicity usually exhibit high fluidity, which facilitates their storage. HP-β-CD microcapsules exhibited high hygroscopicity (Table 1) owing to the many hydroxyl groups of the wall material (HP-β-CD) bonding with the hydrogen atoms of water molecules. The anticaking agents reduced hygroscopicity by preventing the hydroxyl groups of HP-β-CD forming hydrogen bonds with water. A general relationship exists between moisture content and hygroscopicity: microcapsules with a high moisture content show high hygroscopicity and easily absorb moisture from their surroundings.

### 2.3. Powder Flowability of Microcapsules with Different Anticaking Agents

The angle of repose and HR of the powder reflect the flowability of the microcapsules. The angle of repose depends on powder friction; microcapsules with a small angle of repose exhibit low levels of friction and high flowability, which reduce the degree of caking. The angle of repose is classified as follows: 25°–30°, high flowability powder; 30°–38°, medium flowability powder; 38°–45°, low flowability powder; 45°–55°, cohesive powder; and 55°–70°, very cohesive powder [25]. As shown in Table 1, HP-β-CD microcapsules exhibited the highest angle of repose and the anticaking agents reduced this angle, i.e., they improved the flowability of the microcapsules. The smallest angle of repose was measured in HP-β-CD/MAN microcapsules (32.90° ± 0.85°), followed by HP-β-CD/CaSt microcapsules (35.27° ± 0.39°) and HP-β-CD/MgSt microcapsules (36.17° ± 0.99°).

A low HR, i.e., the ratio of the bulk density to the tapped density, reflects the low cohesion and high flowability of microcapsules. We used the following flowability classifications according to HR: 1.00–1.11, excellent flowability; 1.12–1.18, good flowability; 1.19–1.25, fair flowability; 1.26–1.34, passable flowability; 1.35–1.45, poor flowability; and >1.45, very poor flowability [24,26]. The HP-β-CD microcapsules exhibited the highest HR value with very poor flowability, whereas the microcapsules with anticaking agents exhibited reduced HR values to varying degrees (Table 1). HP-β-CD/MAN microcapsules exhibited the lowest HR value (1.33 ± 0.03); thus, MAN as an anticaking agent markedly reduced powder cohesion and improved flowability.

### 2.4. FTIR, Thermal, and XRD Analysis of Microcapsules with Different Anticaking Agents

#### 2.4.1. FTIR Analysis of HP-β-CD Microcapsules

The FTIR spectra of the microcapsule powders are shown in Figure 3. The spectra of HP-β-CD microcapsules and blank microcapsules had peaks at 3400, 2922, 2849, 1743, and 1650 cm^−1^ (Figure 3A). The band observed at 3700–3000 cm^−1^ is characteristic of the-OH group, the bands at 2922 and 2849 cm^−1^ are caused by the stretching vibration of the C-H group, and the bands at 1743 and 1650 cm^−1^ can be attributed to the C = O stretching of ester groups. In addition, the spectra obtained for lavender essence (Figure 3B) had bands at 1240, 1088 and 860 cm^−1^. The bands at 1240 cm^−1^ correspond to a stretching vibration of the C = O group of linalyl acetate, those at 1088 cm^−1^ result from the stretching vibration of the C-OH group of linalool, and those at 860 cm^−1^ are associated with the out-of-plane bending vibration of the = CH group on the benzene ring of thymol. All of the abovementioned bands were observed in the FTIR spectra of HP-β-CD microcapsules, suggesting that the lavender essence was successfully encapsulated.

The characteristic bands of the anticaking agents were also observed in the four microcapsule systems with anticaking agents (Figure 3D). In HP-β-CD/MAN microcapsules, the band at 933 cm^−1^ corresponds to the characteristic band of MAN. In contrast, the stretching vibration of the COO- group of CaSt (1420 cm^−1^) and MgSt (1337 cm^−1^) appeared in HP-β-CD/CaSt and HP-β-CD/MgSt microcapsules, respectively. Similarly, the bands of HP-β-CD/SiO_2_ microcapsules at 1110 and 800 cm^−1^ were caused by the antisymmetric stretching vibration of the Si-O-Si group and the symmetric stretching vibration of the Si-O group, respectively. No new bands were generated in any microcapsule, indicating that the lavender essence was encapsulated by the wall material via a physical interaction, whereas no chemical interaction occurred between the various anticaking agents and HP-β-CD.

#### 2.4.2. Thermal Analysis of Microcapsules with Different Anticaking Agents

Lavender essence was decomposed rapidly and released quickly by heat and its weight was reduced to zero at about 140 °C (Figure 4A), indicating that it could not be processed at high temperature. However, the prepared microcapsules protected the volatilisation of the lavender essence; as the temperature increased, the lavender essence was slowly released from the pores of the microcapsules. When the decomposition temperature of the wall material was reached, the lavender essence was released as the wall material decomposed. Based on the weight loss of microcapsules as temperature was increased, the lavender essence loading capacities of HP-β-CD, HP-β-CD/MAN, HP-β-CD/CaSt, HP-β-CD/MgSt, and HP-β-CD/SiO_2_ microcapsules were calculated as 14.43%, 14.42%, 14.75%, 12.99%, and 10.83%, respectively.

The thermal curves of the four microcapsule systems with anticaking agents were similar, with the three stages of mass loss shown clearly in Figure 4B. The first stage occurred at 30 °C–100 °C, corresponding to the volatilisation of free water molecules. The second stage occurred at 100 °C–230 °C, corresponding to the volatilisation of water molecules in the HP-β-CD cavity and the incompletely embedded lavender essence. The third stage occurred at 230 °C–500 °C, and the quality loss in this stage was mainly due to the decomposition of the microcapsules. These results confirm that the microcapsules improved the thermal stability of the lavender essence.

According to the differential scanning calorimeter patterns (Appendix A), HP-β-CD/MAN microcapsules did not exhibit a glass transition temperature; however, the glass transition temperature of microcapsules with other anticaking agents was 52.62 °C–62.01 °C owing to the different hygroscopicities of the microcapsules. Microcapsules with a high hygroscopicity absorb water spontaneously, decreasing the glass transition temperature. However, when the glass transition temperature is increased, the stability of the microcapsules is also increased. HP-β-CD/MAN microcapsules exhibited an endothermic peak at 238.3 °C but did not exhibit an endothermic peak corresponding to the melting of the raw material (i.e., the endothermic melting peak of raw MAN at 168 °C), indicating that HP-β-CD/MAN microcapsules may not crystallise completely. This result was confirmed by the XRD patterns (Figure 2). In addition, HP-β-CD/MAN microcapsules exhibited multiple endothermic peaks and exothermic peaks at 340 °C–400 °C, indicating that HP-β-CD/MAN microcapsules may undergo a crystalline transformation.

#### 2.4.3. XRD Analysis of Microcapsules with Different Anticaking Agents

As shown in Figure 2A, raw HP-β-CD exhibited a broad peak at 15°–24° without sharp diffraction peaks, reflecting its amorphous structure. The blank microcapsule showed a characteristic sharp peak of emulsifier sucrose fatty acid ester at 21.3°. The broad peak of raw HP-β-CD and the sharp peak of sucrose fatty acid ester were both observed in HP-β-CD, HP-β-CD/CaSt, HP-β-CD/MgSt, and HP-β-CD/SiO_2_ microcapsules (Figure 2B). Similar to the blank microcapsule, these microcapsules had amorphous structures with poor crystallinity. As shown in Figure 2B, HP-β-CD/MAN microcapsules had a crystalline structure, as evidenced by many sharp diffraction peaks. Based on observations under a SEM (Figure 1b), HP-β-CD, HP-β-CD/CaSt, HP-β-CD/MgSt, and HP-β-CD/SiO_2_ microcapsules possessed relatively smooth surfaces, whereas HP-β-CD/MAN microcapsules possessed a rough surface due to MAN crystals. This roughness was likely caused by the rapid crystallisation of MAN in the spray-drying process.

### 2.5. Morphology and Microstructure of HP-β-CD/MAN Microcapsules

Our results indicate that MAN is an effective anticaking agent, and HP-β-CD/MAN microcapsules exhibit high flowability and strong anti-agglomeration properties. Thus, the effect of the MAN ratio on the properties of HP-β-CD/MAN microcapsules was investigated with the aim of optimising this ratio.

The surface roughness and shape of microcapsules differed when the MAN ratios were changed (Figure 5). HP-β-CD/MAN microcapsules with a 0% MAN ratio exhibited the agglomeration phenomenon, although agglomeration was relatively loose and the bond between microcapsules was not tight. However, as the MAN ratio was increased, the agglomeration of microcapsule powder was reduced. For example, HP-β-CD/MAN microcapsules with a 12% MAN ratio showed no obvious agglomeration and the powder had high flowability with the form of fine flour. In contrast, when the MAN ratio was increased to 16% or 20%, the microcapsules exhibited obvious ‘hard agglomeration’ with larger particles. The surface of microcapsules with a low MAN ratio appeared to be smooth with large depressions, whereas the surface of microcapsules with a high MAN ratio was completely spherical. In addition, microcapsules with high MAN ratios had rough protruding crystals on the surface, which were likely MAN crystals formed on the surface of the microcapsule shell. Littringer et al. [27] observed a similar surface morphology in spray-dried MAN. MAN also shows a strong tendency to follow the diffusion of water outside the microcapsules and be retained on the microcapsule shells to form crystals after the drying process [28]. Adhesion between microcapsules with high MAN ratios also occurred owing to their rough surfaces. Overall, observations of the powder state and microscopic morphology indicated that HP-β-CD/MAN microcapsules with a 12% MAN ratio exhibited the lowest level of agglomeration.

Microcapsules with 0–20% MAN ratios had tiny pore walls, and the microcapsules exhibited thicker capsule walls and lower hollow volumes as the MAN ratio increased. During spray drying, the droplet moisture evaporates under the high temperature of the drying chamber and a hard shell is formed, although some of the moisture is retained inside the shell. In microcapsules with a low MAN ratio, the moisture inside the microcapsule is vaporised and migrates outside the microcapsule through the pores of the shell. Therefore, a large hollow structure is formed inside the microcapsule and depressions are formed on the microcapsule surface. However, as the MAN ratio is increased, the droplets slowly form a shell during the drying process and MAN crystals are precipitated on the surface of the microcapsule during shell formation, thereby forming a rough microcapsule surface [29]. Owing to the high concentration of retained emulsion inside the microcapsules, smaller hollow structures are formed after drying and the microcapsules exhibit a thicker shell.

### 2.6. Particle Size Distribution, Moisture Content, and Fluidity of HP-β-CD/MAN Microcapsules

#### 2.6.1. Particle Size Distribution of HP-β-CD/MAN Microcapsules

When the MAN ratio of the microcapsules was 4–12%, the particle size distribution differed only slightly among the microcapsules (Table 2). When the MAN ratio was high, the volume-weighted mean diameter of the microcapsules was also high and widely distributed; thus, the microcapsules underwent more agglomeration. Based on a comparison of particle sizes (Table 2) and observations under SEM (Figure 5), we concluded that MAN causes the surface of the microcapsule to appear crystalline and rough, and rough surfaces are more likely to form agglomerates through liquid bridges when the microcapsules collide with each other.

#### 2.6.2. Moisture Content of HP-β-CD/MAN Microcapsules

The moisture content of the microcapsules was negatively correlated with the MAN ratio of the microcapsules (Table 2). As the MAN ratio increased, the moisture content of the powder decreased from 6.99% to 1.48%. This may be attributable to the hydrogen bonds of HP-β-CD binding easily to water molecules in air and microcapsules with a low MAN ratio exhibiting a high moisture content. MAN was tightly bound to HP-β-CD through intermolecular interactions, effectively hindering the hydrogen bonding of water molecules to HP-β-CD. However, the low moisture content of microcapsules with a high MAN ratio could also be explained by their crystallinity [30]. Although MAN has multiple hydroxyl groups, it also has non-hygroscopic properties. Thus, the crystallisation of MAN on the shell surface reduces the hygroscopicity of the microcapsules.

#### 2.6.3. Hygroscopicity Content of HP-β-CD/MAN Microcapsules

High hygroscopicity is undesirable in microcapsule powder because the microcapsules tend to absorb moisture from the environment, leading to agglomeration. As shown in Table 3, differences in microcapsule hygroscopicity were associated with the MAN ratio. As shown in the SEM images (Figure 5), the crystalline structure of MAN was formed on the surface of the microcapsule shell, preventing the hydrophilic groups of cyclodextrins from binding with moisture and thereby reducing the hygroscopicity of HP-β-CD/MAN microcapsules. The particle size of HP-β-CD/MAN microcapsules differed with their hygroscopicity as the MAN ratio changed from 8% to 20%. Compared with microcapsules with a 20% MAN ratio, those with an 8% MAN ratio had a smaller particle size, resulting in a relatively larger surface area to volume ratio and an increase in the hydrophilic groups of HP-β-CD. Therefore, microcapsules with an 8% MAN ratio exhibited higher hygroscopicity due to the absorption of moisture from the environment. Overall, based on hygroscopicity and particle size, HP-β-CD/MAN microcapsules with a 12% MAN ratio may be the optimal HP-β-CD/MAN microcapsules.

#### 2.6.4. Powder Flowability of HP-β-CD/MAN Microcapsules

As shown in Table 2, the angles of repose of microcapsules with a 0% or 4% MAN ratio were >45°, and the powder flowability of these microcapsules was poor. However, increasing the MAN ratio improved the flowability of HP-β-CD/MAN microcapsules, verifying the intuitive state of the lavender essence microcapsule powder shown in Figure 5.

The HR of microcapsules was negatively correlated with their MAN ratio (Table 2). Specifically, when the MAN ratio was increased, the HR was reduced, and the flowability of the microcapsules improved. Microcapsules with a 0–8% MAN ratio exhibited a HR of >1.45 and very poor flowability. Microcapsules with a 12% MAN ratio exhibited a HR of 1.29 and a passable flowability and microcapsules with 16% or 20% MAN ratios exhibited fair flowability. Overall, these results indicate that MAN improves the flowability of HP-β-CD/MAN microcapsules.

### 2.7. Solubility of HP-β-CD/MAN Microcapsules

Solubility is considered an important parameter for reconstitution quality during powder dissolution [31]. All HP-β-CD/MAN microcapsules were soluble in distilled water. The reconstituted emulsions appeared uniformly milky white without differences in colour or the presence of lumps during dissolution (Figure 6A). The particle size distribution of the emulsions was similar in HP-β-CD/MAN microcapsules with 0–20% MAN ratios; the average particle size was in the range of 90–150 nm (Figure 6C). In addition, the microcapsules with high MAN ratios exhibited high dissolution rates (Figure 6B). This result can be explained as follows: (1) microcapsules with low MAN ratios possessed large internal cavities and exhibited low powder density, so they floated on the water surface and dissolved slowly; (2) microcapsules with high MAN ratios possessed small internal cavities and exhibited high powder density, so these microcapsules sank and dissolved easily; and (3) the surface of the microcapsules with high MAN ratios was rough, increasing the contact area between the microcapsules and the water and thereby increasing the speed at which the powder dissolved. Overall, microcapsules with high MAN ratios dissolved quickly, which is a desirable characteristic as HP-β-CD/MAN microcapsules will be wetted rapidly during dissolution without showing agglomeration.

### 2.8. Thermal, XRD, and FTIR Analyses of HP-β-CD/MAN Microcapsules

#### 2.8.1. Thermal Analysis of HP-β-CD/MAN Microcapsules

The thermal gravimetric analysis curves of a single microcapsule raw material are shown in Figure 7A. HP-β-CD and MAN both showed only a partial thermal curve, with HP-β-CD losing weight at 280 °C–430 °C and MAN losing weight at 230 °C–380 °C. However, almost no weight loss occurred in MAN at <100 °C, reflecting the anti-hygroscopic property of MAN. The thermal gravimetric analysis curves of the microcapsules with different MAN ratios are shown in Figure 7B–G. The thermal stability of the lavender essence was markedly improved in these microcapsules. The loading capacities of HP-β-CD/MAN microcapsules with a 0–20% MAN ratio were 15.22%, 15.43%, 14.59%, 14.42%, 10.36%, and 7.22%, respectively.

#### 2.8.2. XRD Analysis of HP-β-CD/MAN Microcapsules

As shown in Figure 8A, HP-β-CD showed an amorphous structure with a broad peak at 15°–24°. Three crystalline forms of MAN exist: α-, β-, and δ-. α-MAN showed characteristic peaks at 9.4°, 13.8°, and 17.3°; β-mannitol showed characteristic peaks at 10.6°, 14.7°, and 16.8°; and δ-mannitol showed a characteristic peak at 9.7° [32,33]. The sharp peaks at 10.6°, 14.7°, and 16.8° indicated that MAN is composed of β-mannitol (Figure 8A).

The XRD spectra of HP-β-CD/MAN microcapsules with different MAN ratios are shown in Figure 8B. All microcapsules showed the broad peaks of HP-β-CD at 15°–24°. However, the microcapsules with 0% and 4% MAN ratios showed the diffraction peak of surfactant sucrose fatty acid ester at 21.3° but not the characteristic peak of MAN, indicating that these microcapsules were mainly in an amorphous state. This result is consistent with the moisture content and hygroscopicity of the microcapsules (Table 2). In general, amorphous samples are hygroscopic and absorb more moisture throughout storage. As the MAN ratio was increased, the peak at 21.3° was masked by the peak of MAN at 21.0° and the characteristic peak of δ-MAN appeared at 9.7°, indicating that MAN in microcapsules with an 8–20% MAN ratio was mainly in δ-crystalline form. According to a comparison of the intensity of the peak at 9.7°, microcapsules with a high MAN ratio also had a high proportion of δ-MAN. Microcapsules with MAN ratios of 8% and 12% exhibited diffraction peaks at 18.6°, consistent with the raw MAN material. The peak at 18.6° was only slightly shifted to 19.0° when the MAN ratio was increased to 16% or 20%. Variations in the concentration of MAN during spray drying is known to result in microcapsules with different amorphous and crystalline structure contents [34]. In the present study, an increase in MAN content increased the relative crystallinity of the microcapsule powder (Table 2). However, owing to the high relative crystallinity of HP-β-CD/MAN microcapsules with MAN ratios of 16% and 20%, these microcapsules developed undesirable ‘hard agglomeration’. Based on a comparison of our XRD (Figure 2 and Figure 8) and SEM (Figure 5) results, we conclude that MAN tends to migrate to the surface of HP-β-CD/MAN microcapsules during spray drying to form a rough crystalline structure on the microcapsule surface (Figure 9).

#### 2.8.3. FTIR Analysis of HP-β-CD/MAN Microcapsules

The spectra of HP-β-CD/MAN microcapsules had peaks at 3400, 2922, 2849, 1743, 1650, 1240, 1088, and 860 cm^−1^ (Figure 10B), indicating that the lavender essence was successfully encapsulated in these microcapsules. In addition, bands at 1160 and 1030 cm^−1^ resulting from the vibration of O-C-O of linalyl acetate and the stretching vibration of C-OH of MAN, respectively, were observed in HP-β-CD/MAN microcapsules. In the FTIR spectrum of raw MAN (Figure 10A), the band at 929 cm^−1^ is characteristic of β-mannitol, indicating that raw mannitol is composed of β-mannitol. As shown in Figure 10C, the FTIR spectra of HP-β-CD/MAN microcapsules with low MAN ratios did not show the characteristic bands of MAN, whereas the band at 967 cm^−1^ in HP-β-CD/MAN microcapsules with MAN ratios of 16% and 20% is due to δ-MAN. This result is consistent with the XRD pattern, indicating that MAN has a crystal form (Figure 8B).

## 3. Materials and Methods

### 3.1. Materials

Food-grade HP-β-CD was purchased from Shandong Binzhou Zhiyuan Biotechnology Co., Ltd., China. MAN was purchased from Aladdin Reagent Co., Ltd., China. CaSt was purchased from Shanghai Titan Technology Co., Ltd., China. MgSt and glyceryl monostearate were purchased from Sinopharm Chemical Reagent Co., Ltd., China. Nano silica was purchased from Nanjing Xianfeng Nano Material Technology Co., Ltd., China. Sucrose fatty acid ester was purchased from Shanghai Yuanye Biological Co., Ltd., China. Lavender essence was formulated in our laboratory.

### 3.2. Preparation of Microcapsule Emulsion and Spray-Drying Conditions

Glyceryl monostearate and sucrose fatty acid esters were mixed with a weight ratio (*w*/*w*) of 1:4 to obtain a mixed surfactant solution, which was dissolved in distilled water at 70 °C and then cooled to ambient temperature (25 °C). HP-β-CD and the various anticaking agents were dissolved in distilled water. The ratio of each wall material combination is displayed in Table 3 and Table 4. The wall material combination and surfactant solution were mixed at 3:1 (*w*/*w*) and stirred with a magnetic stirrer at 800 rpm for 10 min. The lavender essence, which was developed and blended in our laboratory, was then added to this mixture, which was stirred for 0.5 h at 800 rpm; thus, the microcapsule emulsion was prepared [35]. The total solid concentration of the resultant microcapsule was 20% (*w*/*w*).

The prepared emulsions were spray-dried using a SD-06 Spray Dryer (Labplant, North Yorkshire, UK). The spray-dried powders were placed in a collection bottle and stored in a dryer at 25 °C for further evaluation. The following operation parameters were used for spray drying: inlet air temperature, 180 °C; outlet air temperature, 100 °C; feed flow rate, 280 mL/h; inlet air flow rate, 3.5 m/s; and spray nozzle diameter, 0.5 mm.

### 3.3. Characterisation of Microcapsule Powder

#### 3.3.1. Process Yield

The process yield of the microencapsulated powder was calculated as the ratio of the weight of the powder obtained to the solid weight of the feeding emulsion.

#### 3.3.2. Surface Morphology and Internal Structure of the Microcapsules

The morphology of the microcapsule powder was observed using a scanning electron microscope (SEM; Gemini 300, Zeiss, Tokyo, Japan). The microcapsule powder was set on the sample stage and cut using a stainless-steel surgical blade (model J11030 4#). The microcapsules were then coated with a thin layer of gold. The SEM was operated under an acceleration voltage of 0.02–30.00 kV. Accordingly, the surface morphology and internal structure of the microcapsules were observed.

#### 3.3.3. Particle Size Distribution of Microcapsule Emulsion and Powder

The particle size of the microcapsule emulsion was determined using a light scattering apparatus (NanoBrook Omni; Brookhaven, GA, USA). The prepared microcapsule emulsion was diluted 1000 times with deionised water, placed in the sample cell of the instrument, and tested at 25 °C.

The particle size distribution of the microcapsule powder was measured using a laser light diffraction particle-size analyser (Mastersizer 3000; Malvern Instruments, Malvern, UK). The refractive index and test temperature were 1.52 and 25 °C, respectively. The particle size was expressed as the volume-weighted mean diameter (D [3,4]).

#### 3.3.4. Moisture Content and Hygroscopicity of Microcapsule Powder

The moisture content of 3 g of microcapsule powder was determined using an infrared moisture balance (MB27ZH; Ohaus, NJ, USA) at 105 °C. The experiment was repeated three times.

Hygroscopicity was determined according to the method of Santana et al. [36] with some modifications. The microencapsulated powder (1 g) was placed in a desiccator after 24 h prior humidity equilibration with saturated sodium chloride solution and, to ensure constant ambient humidity and temperature, it was placed in a climatic chamber (HHWS-III-150; Shanghai Yuejin Co., Ltd., Shanghai, China) at 25 °C for 12 h. It is worth mentioning that the relative humidity controlled by the saturated sodium chloride solution was 75.29%. The samples were accurately weighed every 1 h. The hygroscopicity was determined as g of absorbed moisture per 100 g dry matter (g/100 g).

#### 3.3.5. Fourier Transform Infrared Spectroscopy Analysis

The chemical structure of the microcapsule powder was determined using a Fourier transform infrared (FTIR) spectrophotometer (Nicolet iN10; Thermo Fisher Scientific, Waltham, MA, USA). The powder was mixed with potassium bromide in an agate mortar and placed in a high-pressure tableting machine to form a pellet. Subsequently, the FTIR spectra were measured in a 4000–500 cm^−1^ wavelength range, and 64 scans with a resolution of 4 cm^−1^ were averaged for each sample.

#### 3.3.6. Thermal Analysis

The thermal properties and glass transition temperatures of the microcapsules were analysed using a SDT Q600 Thermal Analyser (TA Instruments, New Castle, DE, USA). Microcapsules (3.5 mg) were heated on an aluminium pan at 10 °C/min from 30 °C to 600 °C [12,14]. Nitrogen was applied as the protective gas. In order to prevent excessive airflow from blowing away the sample, the flow rate was set at 20 mL/min.

#### 3.3.7. X-ray Diffraction (XRD) Measurement

Relative crystallinity was measured using an X-ray diffractometer (Ultima IV; Rigaku, Tokyo, Japan) [37] with the following scanning conditions: a 2θ range of 5°–90° and a step size of 0.02°. The XRD patterns of the microcapsule powder were collected, and the relative crystallinity was calculated according to Equation (1).
Crystallinity (%) = crystalline area/(crystalline area + amor phous area) × 100(1)

#### 3.3.8. Flowability

The fluidity of a powder is its ability to flow freely in a regular and constant manner, usually expressed by the angle of repose and the Hausner ratio, which is determined by tapping density and bulk density [38]. Microcapsules with a small angle of repose have a high flowability. First, a fixed funnel was kept vertical with the funnel mouth placed 10 cm away from a desktop. Subsequently, 4 g of microcapsule powder was slowly poured into the funnel at a uniform speed and the radius of the cone bottom formed by the microcapsule powder was measured.

To measure bulk density (*ρ_b_*) and tapped density (*ρ_t_*), the microcapsules were first weighed and then poured into a 10 mL graduated cylinder without applying any external force. Bulk density was calculated by dividing the mass (g) of the microcapsule by the graduated cylinder volume. The graduated cylinder was vibrated at a fixed frequency for 5 min and the volume (*v*) and mass (*m*) after this vibration period were recorded to calculate the tapped density of the microcapsules. The calculations were performed in Equation (2). The Hausner ratio (*HR*) was calculated by dividing the tapped density by the bulk density, as shown in Equation (3). All measurements were repeated in triplicate.
*ρ_t_* = *v*/*m*(2)
*HR* = *ρ_t_*/*ρ_b_*(3)

#### 3.3.9. Solubility

To measure solubility, 1 g of microcapsule powder was dissolved in 10 mL of distilled water at ambient temperature (25 °C) and stirred with a magnetic stirrer at 300 rpm. The colour of the reconstituted emulsion was recorded, and the particle size distribution of the emulsion was measured. In addition, the time required to completely dissolve the microcapsule powder was recorded.

### 3.4. Statistical Analysis

All data were analysed using SPSS for Windows (Version 21.0; SPSS inc., Chicago, IL, USA). The reported results are the number-averaged data and standard deviation. *p* < 0.05 was considered statistically significant.

## 4. Conclusions

This study showed that MAN was the superior anticaking agent among those tested, including CaSt, MgSt, and SiO_2_. Microcapsules with anticaking agents exhibited a lower moisture content, hygroscopicity, angle of repose, and HR than HP-β-CD microcapsules without a caking agent. CaSt, MgSt, and SiO_2_ reduced hygroscopicity and caking by increasing the glass transition temperature of the microcapsules. In contrast, the anticaking effect of MAN can be attributed to the hygroscopicity of HP-β-CD being hindered by the crystallisation of MAN. In addition, microcapsules with higher MAN ratios showed lower hygroscopicity and higher powder flowability. However, MAN ratios of 16% and 20% caused the undesirable ‘hard agglomeration’ of HP-β-CD/MAN microcapsules. Considering all tested properties, a 12% MAN ratio provided the optimal HP-β-CD/MAN microcapsule with high powder flowability and low hygroscopicity, reflecting excellent anticaking properties. Notably, the microcapsules exhibited a special Xanthium-type morphology. During spray drying, the concentration of MAN and HP-β-CD increased continuously with moisture evaporation. Given the higher solubility of HP-β-CD, it tended to migrate to the inner layer of the droplet. Conversely, the relatively low solubility of MAN led to its migration to the outer layer of the droplet, where it formed crystals as the concentration became saturated. Thus, the crystallisation of MAN hindered the agglomeration of microcapsules owing to the moisture absorption of HP-β-CD.

## Figures and Tables

**Figure 1 molecules-28-01119-f001:**
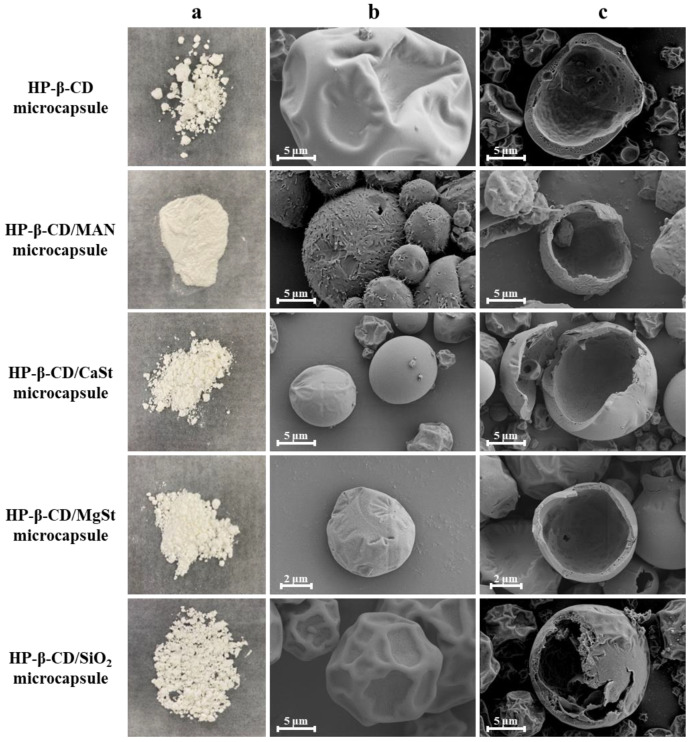
(**a**) Appearance, (**b**) surface morphology, and (**c**) internal structure of microcapsules with different anticaking agents.

**Figure 2 molecules-28-01119-f002:**
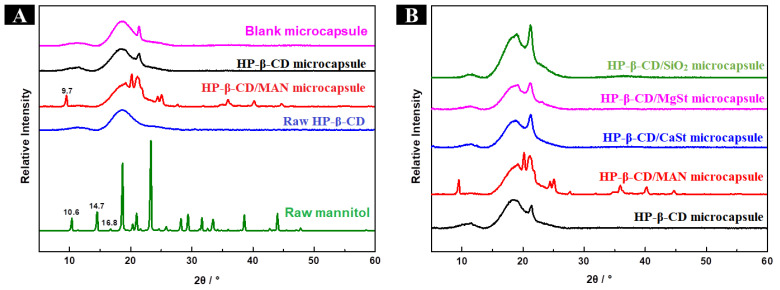
XRD patterns of (**A**) raw materials (HP-β-CD and mannitol), HP-β-CD microcapsules, HP-β-CD/MAN microcapsules, blank microcapsules (without essence) and (**B**) microcapsules with different anticaking agents.

**Figure 3 molecules-28-01119-f003:**
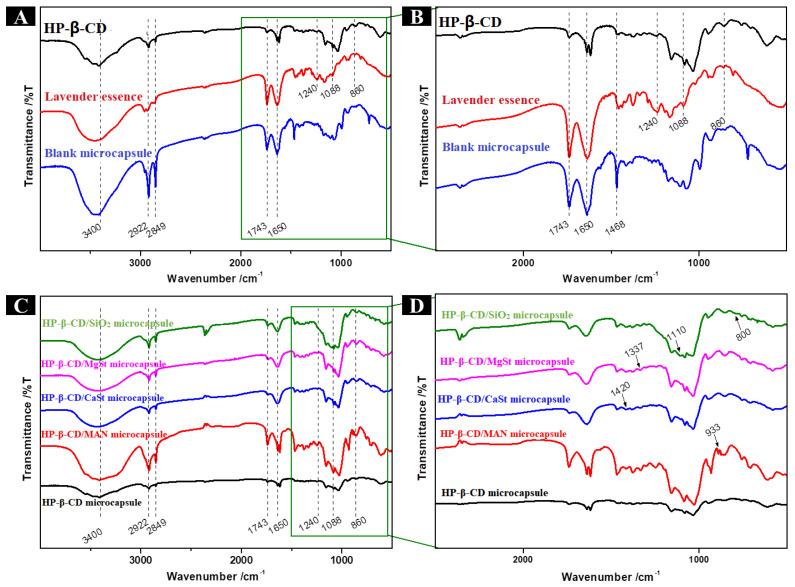
(**A**) FTIR spectra and (**B**) enlarged partial FTIR spectra of HP-β-CD microcapsules, blank microcapsules, and lavender essence. (**C**) FTIR spectra and (**D**) enlarged partial FTIR spectra of microcapsules with different anticaking agents.

**Figure 4 molecules-28-01119-f004:**
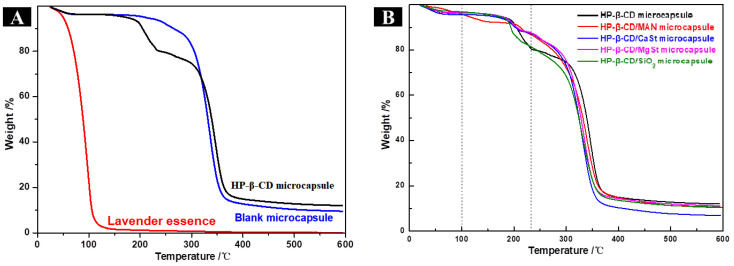
Thermal analysis of microcapsules with different anticaking agents. (**A**) Thermal gravimetric analysis (TGA) patterns of lavender essence, HP-β-CD microcapsules, and blank microcapsules. (**B**) TGA patterns of microcapsules with anticaking agents.

**Figure 5 molecules-28-01119-f005:**
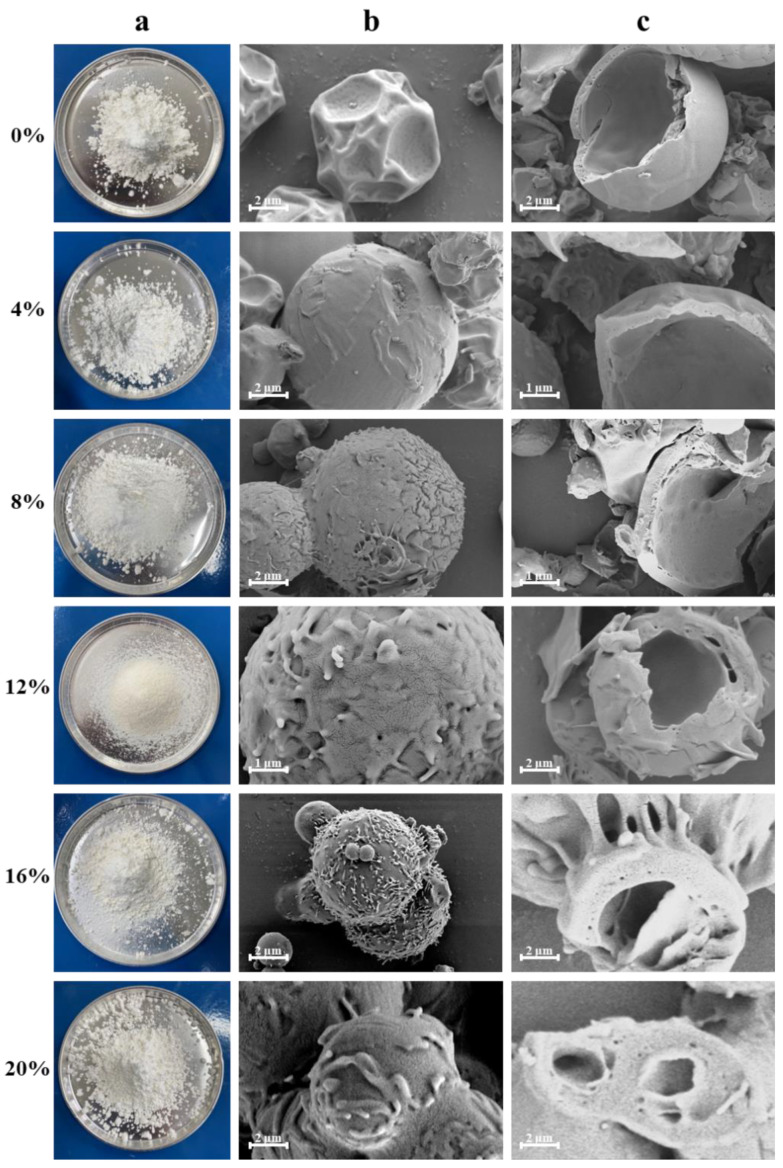
(**a**) Appearance, (**b**) surface morphology, and (**c**) internal structure of HP-β-CD/MAN microcapsules. HP-β-CD/MAN microcapsules with different MAN ratios are shown in different rows, as indicated.

**Figure 6 molecules-28-01119-f006:**
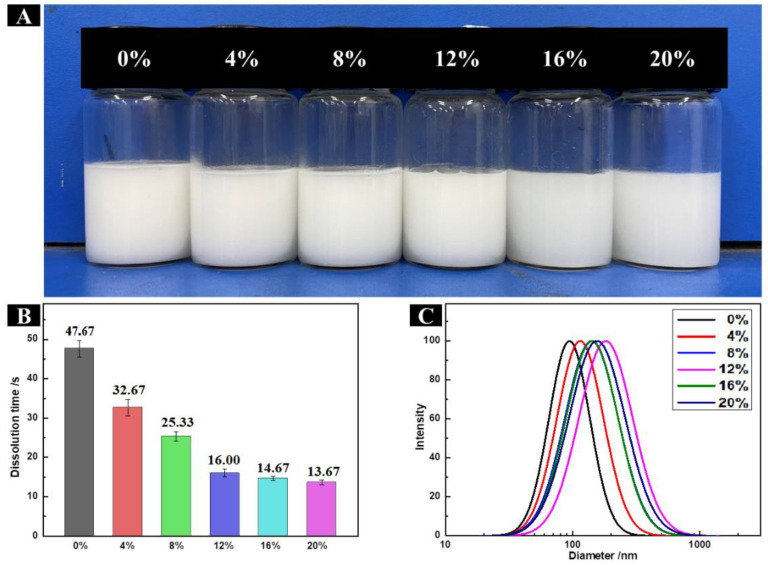
(**A**) Appearance, (**B**) dissolution time, and (**C**) particle size distribution of HP-β-CD/MAN-microcapsule-reconstituted emulsion with different MAN ratios as indicated.

**Figure 7 molecules-28-01119-f007:**
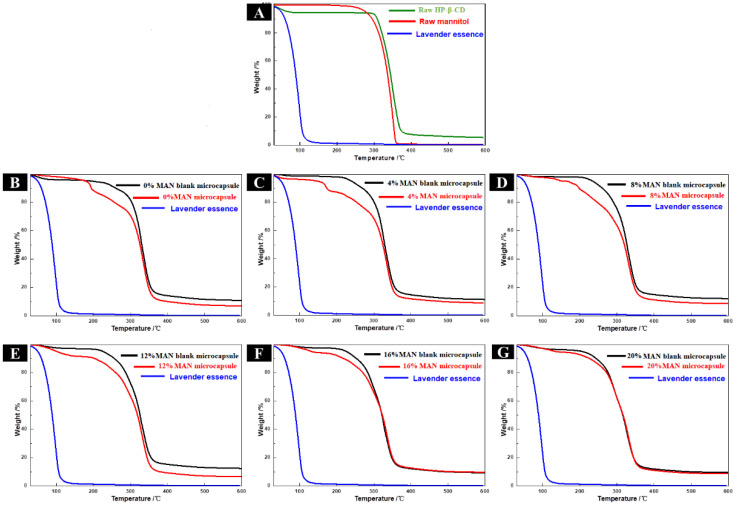
Thermal gravimetric analysis curves of (**A**) raw materials and (**B**–**G**) HP-β-CD/MAN microcapsules. (**B**) to (**G**) show results for HP-β-CD/MAN microcapsules with 0%, 4%, 8%, 12%, 16%, and 20% MAN ratios, respectively.

**Figure 8 molecules-28-01119-f008:**
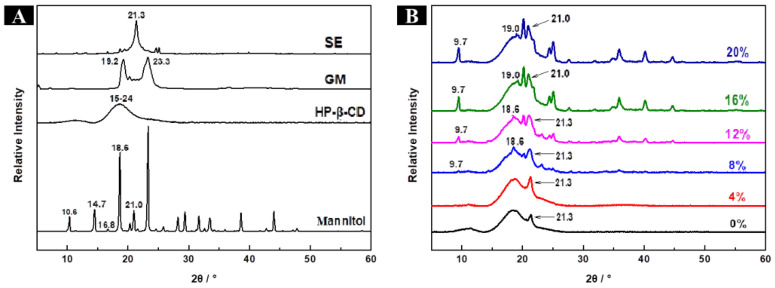
XRD patterns of (**A**) raw materials (HP-β-CD, mannitol, glyceryl monostearate, and sugar esters) and (**B**) HP-β-CD/MAN microcapsules with 0–20% MAN ratios.

**Figure 9 molecules-28-01119-f009:**
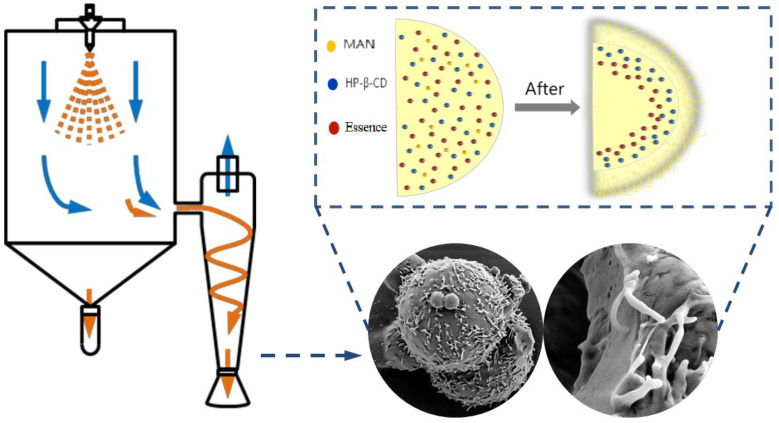
Morphology of HP-β-CD/MAN microcapsules during spray drying.

**Figure 10 molecules-28-01119-f010:**
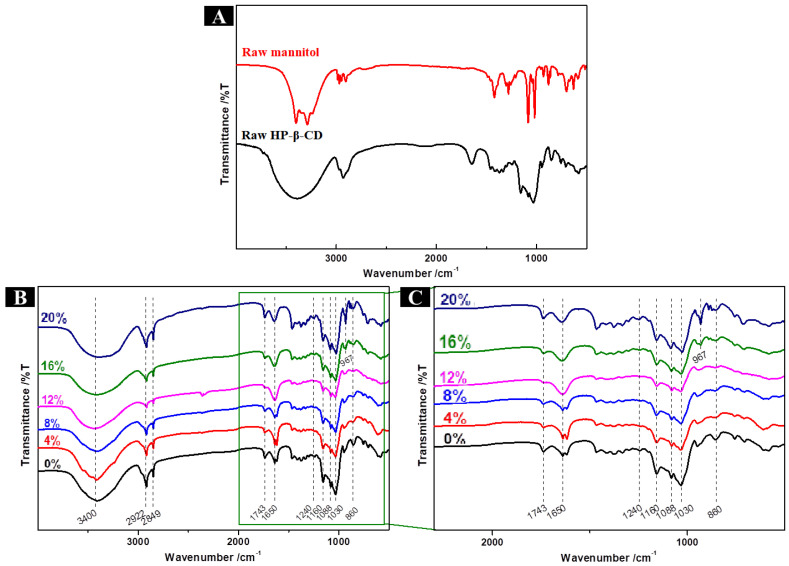
(**A**) FTIR spectra of microcapsule shell materials; (**B**) FTIR spectra and (**C**) enlarged partial FTIR spectra of HP-β-CD/MAN microcapsules with 0–20% MAN ratios.

**Table 1 molecules-28-01119-t001:** Chemical properties of microcapsules with different anticaking agents.

Microcapsule Formulation	Yield (%)	Particle Size of Emulsion (nm)	Particle Size of Powder (μm)	Moisture Content (%)	Hygroscopicity (%)	Hausner Ratio	Angle of Repose (°)
HP-β-CD	50.65 ± 2.48 ^a^	212.80 ± 3.76 ^a^	8.90 ± 0.29 ^b^	6.99 ± 0.14 ^a^	4.38 ± 0.04 ^a^	2.21 ± 0.07 ^a^	49.68 ± 1.00 ^a^
HP-β-CD/MAN	50.80 ± 1.75 ^a^	197.62 ± 4.77 ^b^	11.13 ± 0.38 ^a^	4.03 ± 0.12 ^b^	3.97 ± 0.10 ^c^	1.33 ± 0.03 ^d^	32.90 ± 0.85 ^d^
HP-β-CD/CaSt	37.30 ± 3.62 ^c^	180.49 ± 4.22 ^d^	11.53 ± 2.31 ^a^	3.43 ± 0.11 ^c^	2.97 ± 0.05 ^d^	1.42 ± 0.06 ^b^	35.27 ± 0.39 ^c^
HP-β-CD/MgSt	40.20 ± 2.97 ^b^	194.26 ± 3.31 ^b^	11.83 ± 2.00 ^a^	3.56 ± 0.05 ^c^	3.50 ± 0.04 ^c^	1.43 ± 0.06 ^b^	36.17 ± 0.99 ^c^
HP-β-CD/SiO_2_	49.27 ± 1.09 ^a^	186.36 ± 2.23 ^c^	8.21 ± 0.58 ^b^	4.55 ± 0.14 ^b^	4.05 ± 0.12 ^ab^	1.72 ± 0.10 ^ab^	41.97 ± 0.29 ^b^

Different lower-case letters within the same column mean significant difference at 0.05 level.

**Table 2 molecules-28-01119-t002:** Chemical properties of HP-β-CD/MAN microcapsules with different MAN ratios.

MAN Ratio (%)	Yield (%)	Particle Size of Emulsion (nm)	Particle Size of Powder (μm)	Moisture Content (%)	Hygroscopicity (%)	Hausner Ratio	Angle of Repose (°)	Relative crystallinity (%)
0	49.22 ± 4.35 ^ab^	219.10 ± 3.70 ^a^	8.78 ± 1.06 ^c^	6.99 ± 0.20 ^a^	5.34 ± 0.31 ^a^	2.21 ± 0.07 ^a^	49.54 ± 1.00 ^a^	15.83 ^e^
4	50.57 ± 0.90 ^a^	210.45 ± 3.71 ^bc^	8.53 ± 1.07 ^c^	3.21 ± 0.12 ^b^	5.29 ± 0.13 ^a^	1.78 ± 0.02 ^b^	48.58 ± 0.50 ^a^	33.43 ^d^
8	48.55 ± 2.14 ^b^	212.16 ± 3.55 ^b^	7.17 ± 0.65 ^cd^	2.55 ± 0.10 ^c^	4.37 ± 0.31 ^b^	1.75 ± 0.05 ^b^	44.54 ± 0.96 ^ab^	51.97 ^c^
12	51.14 ± 2.17 ^a^	207.72 ± 5.73 ^c^	8.00 ± 0.45 ^cd^	2.47 ± 0.18 ^c^	3.99 ± 0.30 ^c^	1.29 ± 0.02 ^c^	33.65 ± 0.45 ^b^	64.18 ^b^
16	33.92 ± 2.31 ^b^	212.82 ± 2.25 ^b^	24.57 ± 3.64 ^b^	2.31 ± 0.07 ^c^	3.68 ± 0.16 ^bc^	1.24 ± 0.08 ^c^	22.62 ± 1.29 ^c^	66.20 ^a^
20	19.79 ± 2.29 ^d^	213.71 ± 2.17 ^b^	57.67 ± 5.39 ^a^	1.48 ± 0.11 ^d^	3.23 ± 0.31 ^d^	1.20 ± 0.03 ^c^	21.25 ± 0.81 ^d^	67.43 ^a^

Different lower-case letters within the same column mean significant difference at 0.05 level.

**Table 3 molecules-28-01119-t003:** Wall material ratio of hydroxypropyl-β-cyclodextrin (HP-β-CD)/mannitol (MAN) microcapsules with different MAN ratios.

MAN Ratio of Microcapsules	Anticaking Agent	HP-β-CD (% *w*/*w*)	Mannitol (% *w*/*w*)
0	Mannitol	30	0
4	Mannitol	28	2
8	Mannitol	26	4
12	Mannitol	24	6
16	Mannitol	22	8
20	Mannitol	20	10

**Table 4 molecules-28-01119-t004:** Wall material ratio of microcapsules with different anticaking agents.

Microcapsule Formulation	Anticaking Agent	HP-β-CD (% *w*/*w*)	Anticaking Agent (% *w*/*w*)
HP-β-CD microcapsule	None	5	1
HP-β-CD/MAN microcapsule	Mannitol	5	1
HP-β-CD/CaSt microcapsule	Calcium stearate	5	1
HP-β-CD/MgSt microcapsule	Magnesium stearate	5	1
HP-β-CD/SiO_2_ microcapsule	Silica	5	1

## Data Availability

Not applicable to this article.

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
