# Peer review of "Mannitol Is a Good Anticaking Agent for Spray-Dried Hydroxypropyl-Beta-Cyclodextrin Microcapsules"

_molecules, 2023, doi:10.3390/molecules28031119_

Round 1
Reviewer 1 Report
This article investigated the effect of different anti-caking agents on the physicochemical properties of microcapsules prepared by spray drying method, and the study is abundant, but there are problems that the structure of the article is not clear enough and the analysis of the research results is not clear enough. There are many grammatical and writing errors in the article. In addition, some references in the article are not properly cited. My detailed comments are as follows:
1. The title is the study of different anti-caking agents, in order to improve the structure of article, you should study either the effect of different anti-caking agents on the same concentration, or different concentrations of different caking agents.
2. Line 106 please adjust the order of the anti-caking agent, which is consistent with what you mentioned before.
3. Line 138 Please add the necessary content about the extraction process of lavender essence. If the extraction process has already been reported please add references.
4. Line 173 please modify this sentence.
5. Line 203 please add the flow rate of the protective gasthis parameter. This parament is very important for the determination of the glass transition temperature of the sample, which fails from the this results.
6. Line 212 please delete equations 2 and 3 here.
7. Line 220 FT-IR determination parameters need to be supplemented.
8. Line 268 please delete “ and embedding effect”.
9. Line 302 please delete “microcapsules with different ... indicated”
10. Line 307 What does “blank microcapsules” mean?
11. Line 321 “The particle size ... microcapsules” this is a wrong conclusion
12. Line 347 reference 27, this stduy does not mention this conclusion.
13. Line 348-352. please modify this sentence.
14. The order of results and discussion should be consistent with the methodology, and the content should correspond to each other and analyze the results point by point.
15. The results did not reflect the significance analysis and did not fully exploit the experimental data.
Author Response
请看附件。

Reviewer 2 Report
Comments by Reviewer
Ms. ID. Molecules-2083663
Effects of anticaking agents on spray-dried hy-droxypropyl-beta-cyclodextrin microcapsules
GENERAL COMMENT: The manuscript reports the anticaking effect of different compounds: calcium stearate, magnesium stearate, silicon stearate and mannitol. The results indicate that mannitol has a higher anticaking effect. The authors did additional experiments to determine some features of the anticaking effect of mannitol. The title could be improved and may be include relevant words that indicates the study of the mannitol anticaking effect.
1. Line 1. The manuscript was presented as a “Review”, please correct as a research paper.
2. The location of the figures in the manuscript should be adjusted.
3. All tables should include mean comparative test, ej. Tukey test.
4. The presentation of the results should be improved, due to all experimental test are presented two times, one to show the comparison between different anticaking compounds and the second time show the comparison between different MAN concentrations.
5. Line 144. The title of the table should provide complete names of the chemical compounds and the abbreviature.
6. Lines 193. Indicate the entire process to cut the microcapsules “using a surgical knife blade”, provide de number of the blade and complete reference of the materials.
7. Lines 199- 203: Please improve the method to the measure of glass transition temperature, sample moisture during the experiment and other relevant information. Please provide the reference of the method and heating rate to measure the glass transition temperature. Why the final temperature was 600 °C? Please provide references of the method.
8. Figure 4-C. The figure should be improved, due to the Tg is not possible to identify. The base line of the thermogram is not clear and the thermal transitions are not defined in the graph.
9. Line 226. The authors state “The colour of the reconstituted emulsion was recorded”; it is a solution, emulsion or dispersion of the microcapsules?
10. Line 233. Correct: “Hausner ratio”.
11. Line 270. “The particle size of the microcapsules was mostly”: Please indicate a numerical or statistical value of “mostly”.
12. Line 299. Is the rough of the microcapsules a defect during the spray drying process?
13. Table 3. Table 3 shown the particle size of microcapsules, but there are not differences between the MAN, CaSt, MgSt microcapsules size. Explain how the authors concluded that MAN is the best anticaking, if the particle size is not different. Include the particle size distribution of the microcapsules to support the results.
14. Line 501-502: The authors states: “This roughness was likely caused by the rapid crystallization of MAN in the spray drying process”, please improve discussion, which is the cause of the “rapid crystallization”?
15. Figure 5. The SEM images of the samples 8% and 20% shown agglomeration between two or more microcapsules. How do the authors explain that? Is Mannitol an efficient anticaking?
16. Figure 9. The figure does not show a crystal formation, please improve the title.
Author Response
请看附件。

Round 2
Reviewer 1 Report
The article has been reasonably revised according to the comments of the reviewers.